# Recaptured Raw Screen Image and Video Demoiréing via Channel and Spatial Modulations

**Huanjing Yue**
Tianjin University
Tianjin, China
huanjing.yue@tju.edu.cn

**Yijia Cheng**
Tianjin University
Tianjin, China
yijia_cheng@tju.edu.cn

**Xin Liu**
Tianjin University
Tianjin, China
Lappeenranta-Lahti University of Technology LUT
Lappeenranta, Finland
linuxsino@gmail.com

**Jingyu Yang** [*]
Tianjin University
Tianjin, China
yjy@tju.edu.cn

## Abstract

Capturing screen contents by smartphone cameras has become a common way for information sharing. However, these images and videos are often degraded by moiré patterns, which are caused by frequency aliasing between the camera filter array and digital display grids. We observe that the moiré patterns in raw domain is simpler than those in sRGB domain, and the moiré patterns in raw color channels have different properties. Therefore, we propose an image and video demoiréing network tailored for raw inputs. We introduce a color-separated feature branch, and it is fused with the traditional feature-mixed branch via channel and spatial modulations. Specifically, the channel modulation utilizes modulated color-separated features to enhance the color-mixed features. The spatial modulation utilizes the feature with large receptive field to modulate the feature with small receptive field. In addition, we build the first well-aligned raw video demoiréing (RawVDemoiré) dataset and propose an efficient temporal alignment method by inserting alternating patterns. Experiments demonstrate that our method achieves state-of-the-art performance for both image and video demoriéing. *We have released the code and dataset in https://github.com/tju-chengyijia/VD_raw.*

## 1  Introduction

With the popularity of mobile phone cameras, the way of information transmission has gradually evolved from text to images and videos. However, when capture the contents displayed on digital screens, the recaptured images and videos are heavily influenced by the moiré patterns (i.e., colored stripes, as shown in Fig. 1) due to the frequency aliasing, and the visual contrast are also degraded. Therefore, the task of image and video demoiréing is emerging.

In recent years, image demoiréing has attained growing attention and the demoiréing performance has been greatly improved with the development of deep learning networks and the constructed paired demoiréing datasets. Since the moiré patterns are of various scales and is distinguishable in frequency domain, they are removed by constructing multi-scale fusion networks [19; 6; 28; 14], frequency

---

[*]Corresponding author. This work was supported in part by the National Natural Science Foundation of China under Grant 62072331, Grant 62231018, and Grant 62171309.

37th Conference on Neural Information Processing Systems (NeurIPS 2023).

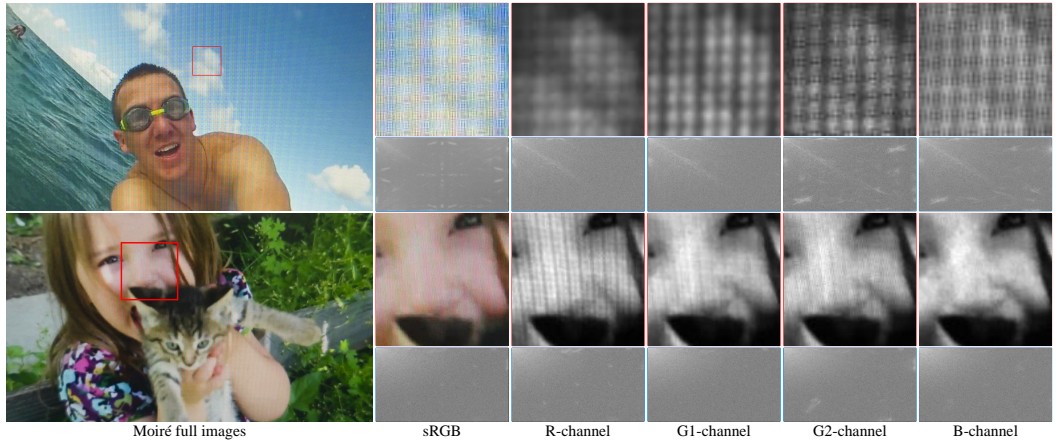

| Moiré full images | sRGB | R-channel | G1-channel | G2-channel | B-channel |

Figure 1: Visualization of moiré patterns in different color channels and their corresponding DCT spectra. For the sRGB channel, we only present the DCT spectrum of its intensity (Y) channel.

domain modeling [11; 41; 42], or moiré pattern classification [5]. These works all perform demoiréing in standard RGB (sRGB) domain. In these works, the moiré patterns are not only caused by frequency aliasing between the camera's color filter array (CFA) and the grids of display screen, but also by the demosaicing (interpolating new pixels, further aggravating frequency aliasing) algorithms in the image signal processing (ISP) [34]. In contrast, the moiré patterns in raw domain is simpler, the raw data contains the most original information, and processing moiré patterns in raw domain is more consistent with the imaging pipeline. Therefore, Yue *et al.* proposed demoiriéing in raw domain [34] and demonstrated the superiority of the raw domain processing over sRGB domain processing. However, they directly processed the raw input with an encoder-decoder network, without exploring the special properties of moiré patterns in raw domain. Therefore, developing a tailored demoiréing method for raw inputs is demanded.

Compared with image demoiréing, video demoiréing is rarely studied. Recently, Dai et al. [3] proposed the first video demoireing network with a temporal consistency loss and constructed a video demoiréing dataset. However, their dataset is constructed with one camera and one screen, which limits the diversity of moiré patterns. In addition, there are no raw videos in this dataset. Meanwhile, we observe that raw video processing is emerging in other low-level vision tasks, such as raw video enhancement [1], denoising [31; 23], and super-resolution [13; 37]. Therefore, developing a raw video demoiréing method is also demanded.

We observe that the moiré patterns in different color channels of the raw inputs have different properties. Fig. 1 presents the moiré patterns from two frames. For each image, the top row visualizes the moire patterns in different color channels of the zoomed in patches and the bottom row presents the DCT (Discrete Cosine Transform) spectra of corresponding channels. The patches are normalized for better observation. It can be observed that the moiré patterns in sRGB domain have complex colors and the scales are different. Meanwhile, the moiré patterns in its raw RGGB channels have different intensities and stripes. For example, for the first image, the moiré patterns in B-channel are more heavier while for the second image, the R-channel has more moiré patterns. This also leads to different DCT spectra, where the bright spots represent the frequencies of moiré patterns (more analysis is provided in the appendix). However, this phenomena does not exist in the sRGB domain since the demosaicing operation mixes these moiré patterns and the green channel always has the least moiré pattern due to large sampling rate. Therefore, how to utilize the above properties for raw image (video) demoiréing is important. In this work, we utilize different weights to modulate the color-separated features from the four color channels. In addition, the period of moiré patterns may be large. Therefore, the features with a large receptive field can modulate the features with a small receptive field. Based on the above observations, we propose to perform raw image and video demoiréing via channel and spatial modulations. Our contributions are as follows.

First, considering that the traditional cross-channel convolution will enhance the shared moiré patterns in different channels, we propose to utilize two branches. One branch is cross-channel convolution for shared feature enhance and the other is color-group convolution with learnable parameters modulating

different color groups. Fusing the two branch features together can help remove moiré patterns and recover shared visual structures.

Second, the moiré patterns usually have a large spatial period. Therefore, we propose to utilize features with large receptive field (realized by depth-wise convolution) to modulate the features with small receptive field. In this way, we can use small computation cost to take advantage of the correlations in a large receptive field.

Third, we construct the first real raw video demoiréing dataset with temporal and spatial aligned pairs. Since the frame rate of raw video capturing is unstable, we propose an efficient temporal alignment method by inserting alternating patterns. Experimental results demonstrate that our method achieves the best results on both raw video and image demoriéing datasets.

## 2 Related Works

### 2.1 Image and Video Demoiréing

Compared with traditional demoriéing methods [17; 18; 26; 9; 27], the convolutional neural network (CNN) based image demoiréing methods have achieved much better performance, especially for the recaptured screen image demoriéing. Since the moiré stripes usually have different scales, many methods propose to fuse the features from different scales together [19; 6; 28; 14] to erase both large scale and small scale moiré stripes. Meanwhile, some methods propose to model the moiré patterns in frequency domain, such as wavelet domain filtering [11] and DCT domain filtering [41; 42]. Wang *et al.* combined the two strategies together by developing a coarse-to-fine dmoiréing network. There are also methods trying to use the classification priors. For example, MopNet [5] removed moiré patterns according to the classification of moiré stripes while Yue *et al.* proposed to remove moire patterns according to the content classification priors [34]. Liu et al. proposed utilize unfocused image to help demoriéing of focused image [12]. Many demoiréing methods also emerged in famous competitions, such as the AIM challenge [30] in ICCV 2019 and NTIRE challenge [29] in CVPR 2020. Real-time demoriéing has also been explored [40]. With the development of transformer, a U-Shaped transformer was proposed to remove moiré patterns [22]. Besides supervised demoireing, unsupervised demoireing is also explored [33]. All these methods are proposed for sRGB image demoiréing except [34], which verified that raw domain demoiréing is more appealing.

Compared with image demoiréing, video demoiréing was rarely studied. Dai *et al.* proposed a video demoiréing method with a relation-based temporal consistency loss [3]. They also constructed a moiré video dataset for supervised learning. However, this dataset only contains one camera and one screen, which limits the diversity of moiré patterns.

Different from them, we propose a raw image and video demoiréing method by taking advantage of the moiré differences among different color channels in raw domain.

### 2.2 Raw Image and Video Processing

Since the raw data is linear with imaging intensity and contains the most original information with wider bit depth, many low-level vision tasks have been explored in raw domain. The representative works include raw image and video denoising [10; 21; 31; 23; 32; 4], raw image and video super-resolution [39; 13; 37], raw image and video enhancement [2; 1], and raw image and video HDR [43; 36], etc. These works have demonstrated that raw domain processing can generate better results than sRGB domain processing. For demoiréing, the raw domain processing is lagging behind, especially for raw video demoiréing. In addition, the existed raw demoiréing method [34] did not tailor the networks according to the properties of raw inputs. In this work, we propose to solve image and video demoiréing in raw domain by utilizing the moiré differences in raw domain.

### 2.3 Demoiréing Datasets

The demoiréing performance heavily depends on the training set. Therefore, many datasets for moiré removal have been proposed, which have become essential resources for researchers to train and evaluate their algorithms. The datasets in sRGB domain include synthetic datasets, *i.e.*, LCDmoire [30] and CFAmoire [29], and real datasets, such as TIP2018 [19], FHDMi [6] and MRBI [35]. Raw

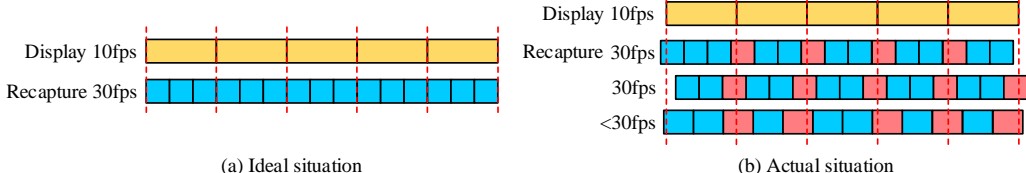

Figure 2: Visualization of the temporal confusion problem during recapturing. (a) is the ideal recapturing situation. (b) shows the actual situation, where the video displaying and recapturing do not have the same start timestamps and the raw domain recording has unstable frame rates. The temporal confusion frames are marked with red boxes.

image demoiréing dataset has also been proposed [34]. During dataset construction, they usually design sophisticated methods to make the recaptured image and the source image be spatial aligned.

Compared with image dataset construction, it is more difficult to construct video demoiréing dataset since it is difficult to keep the displaying and recording be synchronized, which may lead to multi-frame confusion. Dai *et al.* constructed the first video demoiréing dataset [3] but there are many confused frames in their dataset. In addtion, there is no raw videos in this dataset. Therefore, in this work, we construct the first raw video demoiréing dataset by inserting alternating patterns to help select temporal aligned frames from the source and recaptured videos.

## 3 Raw Video Demoiréing Dataset Construction

Although there are many image demoiréing datasets, there's few video demoiréing dataset. In this work, we construct the first raw video demoiréing dataset by capturing the screen with hand-held cameras. We observe that directly capturing the videos displayed on the screen cannot keep synchronization between the displayed frame and recaptured frame due to two problems. First, we cannot exactly make the recording and displaying start timestamps be the same. Second, the camera recording frame rates and video displaying frame rates cannot be exactly matched. Especially for raw video capturing, the recording frame rates are not stable due to the storage speed limits. For example, when we set the recording frame rate to 30 fps, the real recording frame rate varies from 27 to 30 fps. The two problems both lead to multi-frame confusion, as shown in Fig. 2 (b).

To solve the first problem, similar to [3], we insert several white frames at the start and the end of source videos for determining the start and the end of the video. For the second problem, we need to insert a flag on the source frames to help distinguish the confused frames and unconfused frames. A naive flag is expanding the source frames with white or black surrounding areas. In this way, we can select the unconfused frames based on the pixel intensity in the surrounding areas. However, this will cause unconsistent brightness of selected frames. Therefore, we propose to insert slash and backslash patterns alternatively surrounding the source frames, as shown in Fig. 3 (a). As shown in Fig. 3 (b), for confused frames, the patterns surrounding the frame will become crossed patterns since displayed frames switch but they are recorded into one frame. Therefore, we can select the unconfused frames based on the patterns, which is formulated as flag recognition step. Specifically, to reduce the influence of noise and moiré artifacts on pattern recognition, we first perform Gaussion filtering on the original captured frames ($\bar{V}_{\text{raw}}$), then we improve the contrast of the filtering results via a sigmoid function to make the edge clear. Afterwards, we use Canny edge detection to extract edges on the surrounding area and utilize Hough transform to detect the endpoints of the lines, and calculate the average slope of all lines. If the mean slope value is around 1 (-1), it corresponds to the slash (backslash) pattern. Otherwise, if the slope value is around 0, it corresponds to the cross pattern. In this way, we can remove the confused frames with cross pattern. Then we remove the repetitive frames with the same content by calculating the difference of slope, resulting in the selected raw video $V_{\text{raw}}$.

The capturing process, as shown in Fig. 3, can be summarized as follows. Given a set of source frames, we first expand them by alternatively inserting slash and backslash patterns in the surrounding areas. Then, we insert white frames at the beginning and ending of the source frames. For these white frames, we insert a box to help indicate the recording regions in the following frames. The reorganized frames, denoted as $T_{\text{sRGB}}$ are displayed on the screen with a frame rate of 10 fps. Then, we recapture these frames with mobile phones. Since we need to record the raw frames, we utilize

the MotionCam [2] App to help save and export raw frames. The recording frame rate is set to 30 fps. Note that we need to fix the exposure parameters and white balance parameters when shooting, otherwise the color and brightness of recaptured videos will be unstable. The recaptured video frames are denoted as $\bar{V}_{raw}$. Then we select the suitable recaptured frames via deleting the white frames, recognizing the flag pattern, and removing repetitive frames, resulting in $V_{raw}$. In this way, we construct the temporal consistent pairs $(T_{sRGB}, V_{raw})$.

Note that although $T_{sRGB}$ and $V_{raw}$ are temporal aligned, they have spatial displacements. Therefore, for each video pair, we further utilize spatial alignment strategies to make them pixel-wise aligned. Similar to [34], we first utilize homography transform to coarsely align $T_{sRGB}^t$ with $V_{raw}^t$, where $t$ is the frame index. Then, we utilize DeepFlow [24] to perform pixel-wise alignment, resulting in $\hat{T}_{sRGB}^t$. Finally, we crop the center area of $\hat{T}_{sRGB}^t$, and $V_{raw}$, generating the video pairs with a resolution of 720p. Note that, we also export the sRGB version of the recaptured videos, denoted as $V_{sRGB}$. Therefore, we also generate the video demoiréing pairs in sRGB domain, namely $\hat{T}_{sRGB}$ and $V_{sRGB}$.

The displayed source videos are from DAVIS [15], Vimeo[3], high-quality cartoons, and screen recordings, such as scrolling documents, webpages, and GUIs. To ensure the quality of the videos, we select the video whose resolution is ( or larger than) 1080p for recapturing. In this way, our dataset includes natural scenes, documents, GUIs, webpages, etc. We totally generated 300 video pairs and each video contains 60 frames. To ensure the diversity of moiré patterns, we utilize four different combinations of display screens and mobile phones. We change the distances between our camera and the display screen from 10 to 25 centimeters. The view angle changes from 0 to 30 degrees. Please refer to the appendix for more details about our dataset.

Compared with the video demoiréing dataset in [3], our dataset contains more diverse contents, and the moiré patterns are also more diverse than that in [3]. In addition, our dataset has no confusion frames while [3] contains confusion frames since its selecting strategy will fail when the frame rate is unstable. We also provide comparisons with synthesized datasets in the appendix. In summary, our dataset contains raw-RGB and RGB-RGB pairs, which can facilitate moiré removal in both raw and sRGB domains.

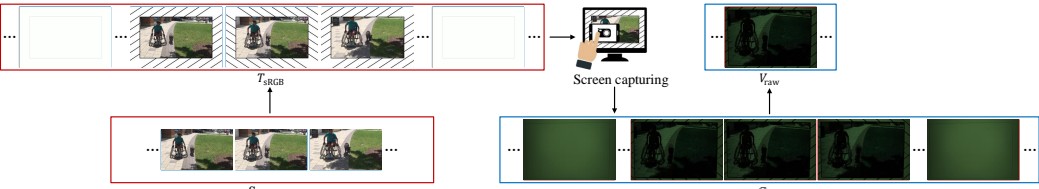

Figure 3: The pipeline of video preprocessing and recapturing.

## 4 Approach

### 4.1 Network Structure Overview

Fig. 4 presents the proposed video and image demoiréing network. For videos, in order to utilize the neighboring frame information, the input is three consecutive frames $\left(V_{raw}^{t-1}, V_{raw}^t, V_{raw}^{t+1}\right)$ and the output is the sRGB result of the central frame $(O_{sRGB}^t)$. The whole pipeline includes preprocessing, feature extraction, alignment, temporal fusion, and reconstruction modules. For the preprocessing, we pack the raw Bayer input with a size of $H \times W$ into four color channels with a size of $H/2 \times W/2 \times 4$. Then, we utilize the normal convolution and group convolution (GConv) to generate color-mixed features and color-separated features (i.e., the features for the R, G1, G2, and B channels are generated separately). Following EDVR [20] and VDmoiré [3], we utilize pyramid cascading deformable (PCD) alignment [20] to align the features of $V_{raw}^{t-1}$ and $V_{raw}^{t+1}$ with those of $V_{raw}^t$. The color-mixed feature and color-separated feature share the same alignment offsets. Then, these features are temporal fused together via a temporal fusion module at multiple scales, which is realized by weighted sum of features from different frames [3]. The fusion weights are predicted from the concatenated temporal features. Afterwards, the fused features go through the reconstruction module to reconstruct the

[2]https://www.motioncamapp.com/
[3]https://vimeo.com/

demoiréing result at multiple scales. Following [3; 25] , the features at different scales are densely connected to make them communicate with each other. Note that, all the operations are applied on two different kinds of features. As shown in the top row of Fig. 4, the first branch is constructed by color-separated features, and the bottom three branches are constructed by color-mixed features. The two kinds of features are fused together at multiple scales with the proposed channel and spatial modulations, which are described in the following. Note that, for image demoiréing, we directly remove the PCD alignment and temporal fusion modules. The remaining modules construct our raw image demoiréing network.

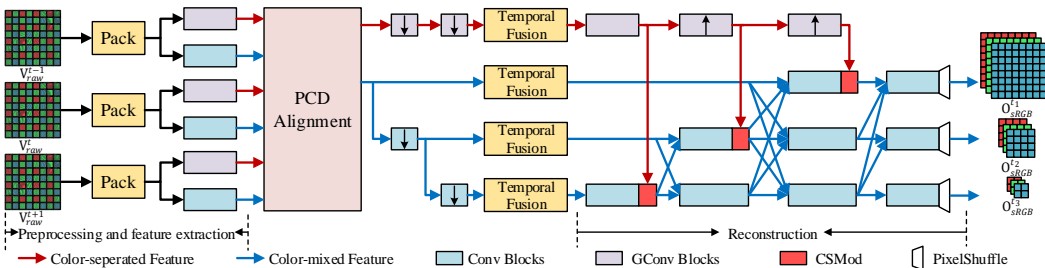

Figure 4: The proposed raw video demoiréing network. The input is three consecutive raw frames $\left(V_{\mathrm{raw}}^{t-1}, V_{\mathrm{raw}}^{t}, V_{\mathrm{raw}}^{t+1}\right)$ and the output is the multi-scale result of the central frame in sRGB domain, and the first scale result $O_{\mathrm{sRGB}}^{t_1}$ is the demoiréing result. The downsampling ($\downarrow$) is realized by strided convolution and upsampling ($\uparrow$) is realized by bilinear interpolation and convolution. For image demoiréing, the PCD alignment and temporal fusion modules are removed.

## 4.2 Channel and Spatial Modulations

In the literature, for image demoiréing, we generally utilize the traditional convolutions to convolve with all channels, generating RGB-mixed (or RGGB-mixed for raw inputs) features. This operator can help recover the shared structures of the three channels. However, this also enhances the shared moiré patterns existed in the three channels. For raw images, we observe that the moiré patterns of the four channels in raw domain are different. As shown in Fig. 1, the moiré intensities in the four channels are different and usually exists one channel with fewer moiré patterns. If we only utilize traditional convolution to recover shared structures, the moiré patterns will also spread to all channels. Therefore, besides the traditional convolution, we further introduce the other branch which utilizes Group Convolution (GConv) to process the packed R, G1, G2, and B channels separately. Since the four groups may contribute differently to the demoiréing process, we further introduce learnable parameters to modulate the four groups (i.e., channel-wise multiplication), as shown in Fig. 5. Then, the color-mixed features and color-separated features are fused together via element-wise addition. In this way, we can combine the advantages of the two features.

Afterwards, we further introduce spatial modulation. The moiré patterns usually have a large spatial period. The receptive field of small convolution filter is limited. Therefore, we propose to utilize the feature with large receptive field to modulate the feature with small receptive field. Inspired by [7], we directly utilize the convolution based modulation to realize this process. As shown in Fig. 5, we first utilize the Linear operation (realized by $1 \times 1$ conv) to transform the fused features. Then, the top branch goes through depth-wise convolution (DConv) with a large kernel ($11 \times 11$), and the generated feature is used to modulate the feature in the bottom branch. Afterwards, the modulated feature further goes through a Linear layer to introduce cross-channel communication which is absent for DConv. After channel and spatial modulation (CSMod), the color-separated feature and color-mixed feature are well fused and enhanced, which are then fed into the following Conv blocks for demoiréing reconstruction.

## 4.3 Loss Functions

We observe that two-stage training is better for our network. Therefore, we utilize a two-stage training strategy. At the first stage, the baseline network (namely removing the color-separated feature and CSMod from Fig. 4) is trained with $\ell 1$ loss and perceptual loss. It is formulated as (for brevity, we

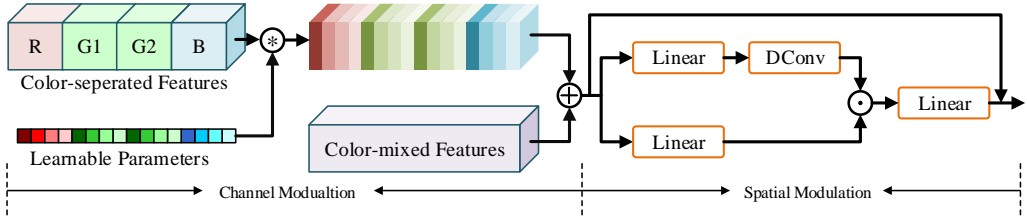

Figure 5: Channel and spatial modulation (CSMod) block.

omit the subscript sRGB for $O^t$ and $\hat{T}^t$ )

$$\mathcal{L}_{s1} = \sum_{i=1}^{i=3} \lambda_1 \|O^{t_i} - \hat{T}^{t_i}\|_1 + \lambda_2 \|\psi_j(O^{t_i}) - \psi_j(\hat{T}^{t_i})\|_1, \tag{1}$$

where $\hat{T}^{t_i}$ is the ground truth and $i$ represents the scale index. $\psi_j$ represents the VGG features extracted at the $j^{th}$ layer, and we utilize the conv1_2 layer in our implementation. $\lambda_1$ and $\lambda_2$ are the weighting parameters. This stage can remove the moiré patterns to some extent but fail for hard samples. For example, if the moiré patterns are strong for one color channel, it is difficult to be removed with only the color-mixed features. Therefore, in the second stage, we introduce the color-separated feature and CSMod to the baseline, and train the whole network. Since we need to recover correct color tones in sRGB domain, similar to [34], we further introduce color loss. Namely, the second stage loss function is

$$\mathcal{L}_{s2} = \sum_{i=1}^{i=3} \lambda_3 \|O^{t_i} - \hat{T}^{t_i}\|_1 + \lambda_4 \|\psi_j(O^{t_i}) - \psi_j(\hat{T}^{t_i})\|_1 + \lambda_5 (1 - \frac{1}{N}\sum_k \frac{O_k^{t_i}\hat{T}_k^{t_i}}{\|O_k^{t_i}\|\|T_k^{t_i}\|}), \tag{2}$$

where $k$ is the pixel index, and $N$ is the pixel number. $\lambda_3$, $\lambda_4$, and $\lambda_5$ are the weighting parameters.

## 5 Experiments

### 5.1 Implementation Details and Datasets

For video-demoiréing, the weighting parameters $\lambda_1$-$\lambda_5$ in Eq. 1 and 2 are set to 0.5, 1, 5, 1, 1, respectively. The first stage training starts with a learning rate of 2e-4, which decreases to 5e-5 at the 37th epoch. The baseline reaches convergence after 40 epochs. In the second stage, the initial learning rate is 2e-4, which decreases to 1e-4 and 2.5e-5 at the 10th and 30th epochs. All the network parameters are optimized by Adam. The batch size is set to 14 and the patch size is set to 256. Image demoriéing shares similar training strategies. The proposed method is implemented in PyTorch and trained with two NVIDIA 3090 GPUs.

For image demoiréing, we utilize the dataset constructed by [34], which contains raw image inputs. For video demoiréing, we utilize the dataset constructed in Sec. 3, which contains 300 video clips. We randomly select 50 video clips to serve as the testing set.

Table 1: Comparison with state-of-the-art image demoiréing methods in terms of average PSNR, SSIM, LPIPS, and computing complexity. The best results are highlighted in bold.

| Method | PSNR↑ | SSIM↑ | LPIPS↓ | Parameters (M) | GFLOPs |
|--------|-------|-------|--------|----------------|--------|
| FHDe$^2$Net | 23.630 | 0.8720 | 0.1540 | 13.59 | 353.37 |
| RDNet | 26.160 | 0.9210 | 0.0910 | 6.045 | 10.06 |
| MBCNN | 26.667 | 0.9272 | 0.0972 | 2.722 | 9.74 |
| MBCNN* | 27.187 | 0.9332 | 0.0888 | 2.717 | 9.67 |
| UHDM | 26.513 | 0.9246 | 0.0911 | 5.934 | 17.56 |
| UHDM* | 27.078 | 0.9302 | 0.0812 | 5.925 | 17.40 |
| **Ours** | **27.261** | **0.9346** | **0.0748** | 5.327 | 21.65 |

Table 2: Comparison with state-of-the-art image and video demoiréing methods for video demoriéing in terms of average PSNR, SSIM, LPIPS, and computing complexity. The best result is highlighted in bold and the second best is underlined.

|  | Method | PSNR↑ | SSIM↑ | LPIPS↓ | Parameters (M) | GFLOPs |
|---|---|---|---|---|---|---|
| **Image** | MBCNN | 25.892 | 0.8939 | 0.1508 | 2.722 | 9.74 |
|  | MBCNN* | 26.082 | 0.8984 | 0.1402 | **2.717** | **9.67** |
|  | UHDM | 26.823 | 0.9078 | 0.1177 | 5.934 | 17.56 |
|  | UHDM* | 25.642 | 0.8792 | 0.1232 | 5.925 | 17.40 |
| **Video** | MFrame-RDNet | 26.970 | 0.9051 | 0.1176 | 6.640 | 34.31 |
|  | MFrame-UNet | 26.185 | 0.8917 | 0.1368 | 32.286 | 39.01 |
|  | VDMoiré | 27.277 | 0.9071 | 0.1044 | 5.836 | 171.93 |
|  | VDMoiré* | 27.747 | 0.9116 | 0.0995 | 5.838 | 43.02 |
|  | EDVR | 26.785 | 0.8993 | 0.1172 | 2.857 | 990.74 |
|  | EDVR* | 27.285 | 0.9064 | 0.1053 | 3.005 | 257.50 |
|  | VRT$^-$ | 24.903 | 0.8728 | 0.1387 | 2.485 | 353.37 |
|  | VRT$^-$* | 27.113 | 0.9034 | 0.1091 | 2.733 | 101.60 |
|  | Ours | **28.706** | **0.9201** | **0.0904** | 6.585 | 70.61 |

## 5.2 Comparison for Image Demoiréing

For image demoiréing, we compare with the raw domain demoiréing method [34] and the state-of-the-art sRGB domain image demoiréing methods, *i.e.*, FHDe$^2$Net [6], MBCNN [41], and UHDM [28]. All these methods are retrained on the raw image dmoiréing dataset [34]. For the most competitive methods MBCNN and UHDM, we further retrain them with raw inputs by revising (changing the input channel number and inserting upsampling layers) the network to make it adapt to raw inputs. The revised versions are denoted as MBCNN* and UHDM*. Since PSNR is heavily affected by the reconstructed color tones, SSIM and LPIPS [38] are good complementary measurements for comprehensive evaluation. As shown in Table 1, our method achieves the best results on all three measurements. Fig. 6 presents the visual comparison results for the top six methods. It can be observed that all the compared methods fail when deal with the regular and strong moiré patterns. In contrast, our method can remove them clearly.

## 5.3 Comparison for Video Demoiréing

To our knowledge, there is only one work, i.e., VDMoiré [3], dealing with video demoiréing. Therefore, we further include other video restoration methods and state-of-the-art image demoiréing methods for comparison. Specifically, we introduce the benchmark CNN based and transformer based video super-resolution networks, i.e., EDVR [20] and VRT [8], for comparison. Since VRT is very large, we reduce its block and channel numbers for a fair comparison, and the revised version is denoted as VRT$^-$. The original loss functions of EDVR and VRT are changed to our loss functions for better demoiréing performance. For image demoiréing, we introduce MBCNN [41] and UHDM [28] for comparison. Since all the compared methods are designed for sRGB inputs, we first retrain them on our RawVDemoiré dataset with sRGB domain pairs. Then, we retrain them with our raw-sRGB pairs. We utilize superscript * to denote the raw input versions. In addition, we transform two raw image processing methods (RDNet [34] and UNet [16]) for video demoiréing by incorporating the PCD module and fusion module. The revised versions are denoted as MFrame-RDNet and MFrame-UNet. The inputs of the two modules are three consecutive raw frames.

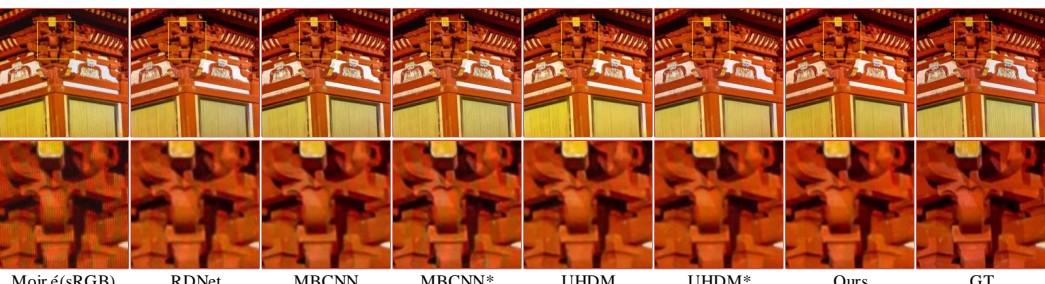

Figure 6: Comparison with state-of-the-art image demoiréing methods.

Table 2 presents the quantitative comparison results. All the raw domain processing methods (except UHDM*) are better than their corresponding sRGB domain versions. All the image based demoiréing methods cannot achieve satisfied results for video demoiréing since they cannot utilize temporal correlations. For EDVR, since it does not reconstruct multiscale results, it cannot remove large moiré patterns. VRT⁻ also cannot generate satisfied results since the shift-window matching mechanism of VRT is not beneficial for large moiré removal. In addition, the revised multi-frame based raw image processing methods MFrame RDNet and MFrame UNet still cannot generate satisfied results. In contrast, our method generates the best demoiréing results, outperforming the second best method by nearly 1 dB. The main reason is that our CSMod strategy can well utilize the complementary information of RGGB channels and can remove strong moiré patterns.

We also give the computing complexity comparison in terms of parameter numbers and GFLOPs in Table 2. It can be observed that the image demoiréing methods consume small computing costs since they process the video frame-by-frame, which also leads to worse demoiréing results. Our GFLOPs are smaller than the popular video restoration frameworks, such as EDVR and VRT, which indicates our method is an efficient video demoiréing method.

Fig. 7 presents the visual comparison results for one video frame. It can be observed that our method can remove the moiré patterns clearly while the compared methods all fail for this hard sample. More results are provided in our appendix.

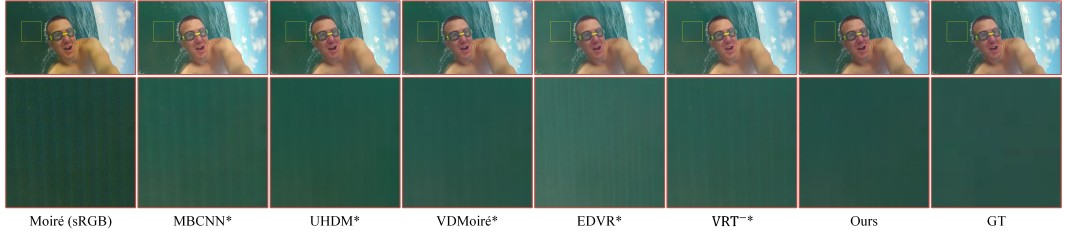

Figure 7: Comparison for video demoiréing.

## 5.4 Ablation Study

Table 3: Ablation study results by adding the proposed modules one by one. The best results are highlighted in bold.

| Variant | PSNR↑ | SSIM↑ | LPIPS↓ |
|---|---|---|---|
| Baseline | 27.963 | 0.9116 | 0.0914 |
| Baseline+SMod | 28.439 | 0.9175 | 0.0939 |
| Baseline+SMod⁺ | 28.498 | 0.9182 | 0.0957 |
| Baseline+CSMod | **28.706** | **0.9201** | **0.0904** |

Table 4: Ablations on loss functions and the multi-scale constraint. The best results are highlighted in bold.

| Variant | PSNR↑ | SSIM↑ | LPIPS↓ |
|---|---|---|---|
| w/o VGG Loss | 28.069 | 0.9166 | 0.1160 |
| w/o Color Loss | 28.642 | 0.9169 | 0.0931 |
| w/o Multi-scale Loss | 27.455 | 0.9124 | 0.1041 |
| Complete | **28.706** | **0.9201** | **0.0904** |

**Ablation on SMod and CSMod.** We perform ablation study by adding the proposed modules one by one. The first variant is our baseline network. The second is introducing the proposed spatial modulation into the baseline network. Namely we only keep the color-mixed features in the channel modulation module in Fig. 5. The third is introducing SMod and replacing the color-separated feature in Fig. 5 with another color-mixed feature (denoted as SMod⁺). The fourth is our complete version, namely Baseline+CSMod. As shown in Table 3, the SMod improves the demoiréing performance by 0.48 dB. One reason is that the SMod can take advantage of large receptive field and the other reason is that we utilize the second stage training loss to update SMod, which is beneficial for color recovery. Compared with SMod, SMod⁺ only achieves little improvement, which demonstrates that the color-separated features are essential. Compared with SMod, the proposed CSMod can further improve 0.26 dB.

**Ablation on Loss Functions.** We further conducted ablation study regarding perceptual loss (VGG Loss), color loss (Color Loss), and multi-scale constraints (Multi-scale Loss), and the results are summarized in Table 4. When the corresponding loss is removed, there is a noticeable decrease in all

evaluation metrics. Perceptual loss places constraints on deep-level features of images, which helps recover structural information, such as shapes and textures. Color loss, on the other hand, focuses more on the global color of images. Recaptured screen images often suffer from weakened attributes like brightness, contrast, color saturation, and details. The constraint provided by L1 loss alone might not be sufficient. Incorporating perceptual loss and color loss can contribute to better image reconstruction quality. Given the multi-scale characteristics of moiré patterns, imposing multi-scale constraints is beneficial for moiré removal and image reconstruction. Note that, the multi-scale constraints are widely utilized in benchmark demoiréing models, such as MBCNN, UHDM, and VDMoiré. In addition, the perceptual loss is also included in all of our compared methods (except MBCNN, which utilizes a more advanced loss function).

## 5.5 Generalization

The image demoiréing experiment is conducted on the dataset provided by [34], whose test set contains two device combinations that are not included in the training set. Therefore, we utilize these test images (about 1/4 of the whole test set) to evaluate the generalization ability of different models. As shown in Table 5, our method still achieves the best results in terms of SSIM and LPIPS, and the PSNR value is only slightly inferior to UHDM*. Note that, for our method, if the test camera-screen combination exists in the training set, the color tone of the test images can be well recovered. Otherwise, the recovered image may have color cast. The main reason is that our network also performs the ISP process, and the ISP process is indeed different for different cameras. Fortunately, the moire patterns can still be removed. Therefore, our LPIPS value is still small. In summary, our method has the best generalization ability in terms of moiré removal.

Table 5: Comparison of the generalization ability of different models by evaluating with the screen-camera combinations that are not included in the training set. The best results are highlighted in bold and the dataset is the image demoriéing dataset [34].

| Method | PSNR↑ | SSIM↑ | LPIPS↓ |
|---|---|---|---|
| FHDe$^2$Net | 22.840 | 0.8968 | 0.1530 |
| RDNet | 24.798 | 0.9280 | 0.0846 |
| MBCNN | 24.726 | 0.9253 | 0.1017 |
| MBCNN* | 24.659 | 0.9314 | 0.0925 |
| UHDM | 24.679 | 0.9239 | 0.0955 |
| UHDM* | **25.769** | 0.9310 | 0.0759 |
| Ours | 25.719 | **0.9333** | **0.0727** |

## 6 Conclusion

In this paper, we propose a novel image and video demoiréing network tailored for raw inputs. We propose an efficient channel modulation block to take advantages of color-separated features and color-mixed features. The spatial modulation module helps our model utilize large receptive field information with fewer computation costs. In addition, we construct a RawVDemoiré dataset to facilitate research in this area. Experimental results demonstrate that our method achieves the best results for both image and video demoiréing.

## 7 Broader Impact

Our work is the first exploring raw video demoiréing and we construct the first RawVDemoiré dataset. To solve the temporal confusion problem during video recapturing, we propose an efficient temporal alignment method, which can be extended to other screen recapturing scenarios. Previous works usually omit the differences between raw and sRGB inputs. Our work is an exploratory work that takes advantage of color-separated features in raw domain and we expect more works along this direction. However, we would like to point out that the demoiréing results may be used for face-spoofing since the moiré detection method cannot identify our result. But on the other hand, our method can imitate the attacks for face recognition, which can help train robust antispoofing methods.

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
