# OpenReview forum: "Recaptured Raw Screen Image and Video Demoiréing via Channel and Spatial Modulations"
_NeurIPS.cc/2023/Conference — NeurIPS 2023 poster_

### Official Review · Reviewer_tgX2 · 2023-06-24

**Soundness:** 3 good
**Presentation:** 2 fair
**Contribution:** 3 good
**Rating:** 6
**Confidence:** 4

**Summary:**

The paper proposed a neural algorithm to remove the moire pattern in an image/video when capturing a display. Since it is observed moire patterns are simpler in raw signals, the proposed method starts from raw data and applies channel and spatial modulations targeting moire pattern properties in the raw domain. A new dataset with better alignment and diverse moire patterns is proposed to empower the network training. The proposed method achieves SOTA scores in the image and video demoireing tasks and promising visual results.

**Strengths:**

* The topic is of broad interest as it is not uncommon to use a cellphone camera to capture contents on the display.
* It is the first paper to explore raw video demoireing.
* The design of the modulations is reasonable. Cross-channel modules in the network better integrate information from 4 raw channels and large receptive fields in cross-spatial modules can handle large moire patterns.
* Compared to the existing video demoireing dataset, the constructed dataset has extra raw signals, more diverse moire patterns and better alignment.
* The proposed method achieves SOTA scores and good visual results.


**Weaknesses:**

* There are existing papers on raw image demoireing [30] and video demoireing [3]. Interestingly, some observations/designs in the paper seem to be consistent with observations/designs in [30] but [30] is not cited in those places, including:
	* finding that the moire pattern in the raw domain is simpler
	* channel-wise modulation through learnable parameters
	* loss function design
The above can considerably diminish the novelty of the paper.

* As the statement of the dataset and train/val/test split is not sufficiently clear (specified in the Question section), I’m unsure about the generalization ability of the proposed method.
* The proposed method doesn't consider the temporal consistency of the demoireing results. This may raise the flickering issue when the method deals with a video with varying moire patterns.


**Questions:**

The authors should do a thorough comparison between this paper and [30] to justify the novelty and highlight the contributions. Also the authors should properly cite [30] wherever necessary.

How does the proposed method trained on the dataset generalizes to an unknown combination of the camera, display and distances? It is possible to justify this using the proposed dataset as it has 4 combinations. For example, the model can be trained on 3 combinations and evaluated on the other one. Unfortunately, I cannot tell how the data is split from the paper. The authors are encouraged to clarify this part, and if the split cannot justify the generalization ability, I suggest an extra experiment on this topic.

Also, how much do distance and viewing angle change? And what is the physical setup of data capture? Do the distance and viewing angle change during the video capture process? A handheld camera usually involves distance and angle changes caused by hand motion, which triggers changes in moire patterns. So it is good to have varying moire patterns in one video. Specifying related details will be helpful for readers to construct their own datasets when necessary.

Why don't make a video dataset using data synthesis approaches as described in [27, 28]? To my opinion data synthesis can greatly diversify the dataset and avoid alignment issues described in Sec 3.


**Limitations:**

* The dataset involves several combinations of displays and cameras, which may be still limited. A model trained on it may bias towards given cameras and display and not reasonably generalize to unseen combinations, unless the authors prove it as mentioned in the Question section.
* As the authors mentioned, the proposed method doesn't consider the temporal consistency of the demoireing results, which may cause the jittering issue.

---

> ### Author Rebuttal · Authors · 2023-08-09
>
> Thanks for your nice summary and positive comments on our contributions and dataset.  In the following, we give detailed responses to your comments. We only give the outline of your questions due to the limited characters.
>
> > About the comparison with [30].
>
> Thanks!  As you mentioned, [30] pointed out that the moiré pattern in raw domain is simpler and [30] utilized hybrid loss functions. We will cite [30] in those places. Note that, the two points are not our contributions in this work. Our main contribution is based on our key observations that the moiré patterns in different raw channels are different. If we utilize traditional convolution to recover shared structures, the moiré patterns in different channels are also spread. Therefore, we propose to use the complementariness between color-separated features and color-mixed features. To our knowledge, there is no work giving this observation and no work exploring the complementariness of color-separated features and color-mixed features for image/video demoiréing.
>
> Specifically, we propose a channel-modulation mechanism to modulate the weights of the color-separated features. If we replace the modulation operation by channel attention, which is used in [30], the performance will be degraded, as shown in Table R2. As shown in Fig. R5, the learned modulation weights have a large variance, which implies that different channels contribute very differently. In contrast, the weights generated by channel attention are similar, which cannot give distinctive enhancements to different channels. Therefore, our modulation strategy is better than the channel attention strategy. We will give a thorough comparison between this work and [30] in our final version.
>
> Table R2: Comparison between our channel modulation and traditional channel attention.
> |Methods|PSNR|SSIM|LPIPS|
> |----|----|----|----|
> |channel attention|28.492|0.9170|0.0987|
> |channel modulation|**28.706**|**0.9201**|**0.0904**|
>
> > About the generalization ability of the proposed method.
>
> Thanks! For video demoiréing, both our training and test datasets include the four device combinations. For each combination, the train and test splits are about 84%/16%. Fortunately, the image demoiréing is conducted on the dataset provided by [30], whose test set contains two device combinations that are not included in the training set. Therefore, we utilize these test images (about 1/4 of the whole test set) to evaluate the generalization ability of different models. As shown in Table R5, our method still achieves the best results in terms of SSIM and LPIPS, and the PSNR value is only slightly inferior to UHDM*. Note that, for our method, if the test camera-screen combination exists in the training set, the color tone of the test images can be well recovered. Otherwise, the recovered image may have color cast. The main reason is that our network also performs the ISP process, and the ISP process is indeed different for different cameras. Fortunately, the moiré patterns can still be removed. Therefore, our LPIPS value is still small. In summary, our method has the best generalization ability in terms of moiré removal.
>
> Table R5: Comparison of the generalization ability of different models by evaluating with the screen-camera combinations that are not included in the training set.
> |Methods|PSNR|SSIM|LPIPS|
> |----|----|----|----|
> |$FHDe^{2}Net$|22.8401|0.8968|0.1530|
> |RDNet|24.7979|0.9280|0.0846|
> |MBCNN|24.7264|0.9253|0.1017|
> |MBCNN*|24.6585|0.9314|0.0925|
> |UHDM|24.6789|0.9239|0.0955|
> |UHDM*|**25.7693**|0.9310|0.0759|
> |Ours|25.7187|**0.9333**|**0.0727**|
>
> >About the dataset capturing details.
>
> Thanks! As you mentioned, using a handheld camera can result in changes in moiré patterns. We intentionally utilized handheld devices to capture videos for the purpose of obtaining varying moiré patterns. In order to capture the videos clearly, we change the distances between our camera and the display screen from 10 to 25 centimeters. The view angle changes from 0 to 30 degrees. Our captured moiré patterns have a large variance in terms of scales, colors, and stripes, as shown in Fig. 2 of the supplementary file.
>
> >Why not using data synthesis approaches [27, 28]?
>
> We would like to point out that there is a significant gap between the synthesized moiré and real moiré patterns. Fig. R6 provides a comparison between [27, 28] and our real dataset. For CFAMoiré [27], it cannot generate moiré patterns in flat areas. However, for recaptured screen images, the flat areas also contain heavy moiré patterns. For LCDMoiré, although it has moiré patterns in flat areas, it cannot synthesize multi-scale interleaving moiré patterns. As shown in Fig. R6, various moiré pattern cycles can be observed on the same image for real captured ones. However, the synthesized moiré patterns lack this feature. In addition, real captured moiré patterns often have diverse colors and morphologies, which are relatively limited in synthesized moiré patterns. Finally, current moiré synthesis methods are all designed for sRGB domain moiré synthesis other than raw domain moiré synthesis. As shown in Fig. R4, even utilizing inverseISP to convert real moiré images into raw domain, we still cannot generate the raw domain moiré patterns that are close to real ones. Therefore, in this work, we captured a raw video demoiréing dataset and generate space and time well aligned pairs to benefit the training and evaluation process.
>
> >About temporal consistency.
>
> Our video demo shows the video moiré removal results and there are slightly jittering artifacts. To solve this problem, we further fine-tune our network by introducing temporal loss [3]. The demoiréing performance and temporal consistence are further improved (see Table R6). We will present the video results in our website.
>
> Table R6:  Ablation on temporal loss.
> |Methods|PSNR|SSIM|LPIPS|
> |----|----|----|----|
> |w/o temporal loss|28.706|0.9201|0.0904|
> |w temporal loss|28.968|0.9200|0.0884|

---

> > ### Comment · Reviewer_tgX2 · 2023-08-17
> >
> > I'm happy about the authors' responses. Admittedly as other reviewers pointed out the improvement of network design is mild, but the idea of raw video demoireing and the dataset capture are of adequate novelty and of broad interest, and the experimental section will be sound after adding additional experiments. Hence I'd like to raise my rating.

---

> > > ### Author Response · Authors · 2023-08-18
> > >
> > > Dear Reviewer,
> > >
> > > Thanks for your time!
> > > Your instructive suggestions have greatly improved our work!
> > >
> > > Best,
> > > Authors

---

### Official Review · Reviewer_piUN · 2023-06-27

**Soundness:** 3 good
**Presentation:** 3 good
**Contribution:** 4 excellent
**Rating:** 6
**Confidence:** 5

**Summary:**

This paper aims at tackle the problem of recaptured raw screen image/video demoiréing. A deep demoireing network with a two-branch structure is proposed, where one branch is a cross-channel convolution for shared feature enhancement and the other is a color group convolution with learnable parameter modulation for different color groups. By fusing the two branch features together, disorder patterns are removed, and the shared visual structure is restored. Furthermore, the paper constructs the first real-world raw video demoireing dataset with temporal and spatial alignment pairs for efficient temporal alignment by inserting alternate patterns. Extensive experiments are conducted to demonstrate the performance of the proposed approach.

**Strengths:**

1. The first raw video demoireing dataset is constructed, which benefits the research of related studies.
2. It provides interesting observations: 1) moiré pattern characteristics are analyzed in the raw domain; 2) moiré pattern intensity is different in different color channels.
3. The data collection technique used in this paper solves the problem of frame misalignment in video collection.
4. The double-branch structure of the proposed network can extract both the color-mixed features and the color-separated features, which capture the moiré features more fully.
5. The experiments are solid and comprehensive.
6. The paper is well written.


**Weaknesses:**

1. The proposed network model does not show an advantage in model size.
2. For image demoireing, the performance gain is relatively small compared to MBCNN*, but the number of references is twice as large.
3. The two-branch structure has no rich interactions, but the network needs two stages for training.
4. The data set produced in this paper includes both raw domain and srgb domain, but only raw domain is used in the experiment, while most of the compared methods are designed based on the SRGB domain.


**Questions:**

Could the authors provide the results and comparison on the SRGB domain?

**Limitations:**

As discussed in the paper, like existing methods, the temporal consistency is not introduced in the proposed one.

---

> ### Author Rebuttal · Authors · 2023-08-09
>
> Thanks for your nice summary and positive comments on our contributions and datasets. In the following, we give detailed responses to your comments.
>
> > The proposed network model does not show an advantage in model size.
>
> We agree with you that our method does not have advantage in model size. In real cases, the inference speed and computing costs are more important metrics since a small model size (the model sizes in our comparisons are all smaller than 14M) does not influence the inference speed and power dissipation. As shown in Table 2, our method demonstrates a lower GFLOPs compared to other video demoiréing methods.
>
> > For image demoiréing, the performance gain is relatively small compared to MBCNN*, but the number of references is twice as large.
>
> Thanks. We agree with you that our model is larger than MBCNN* and PSNR gain over MBCNN* is small. However, for image demoiréing, the LPIPS metric is more sensitive in measuring the quality of moiré removal. As shown in Table 1, our LPIPS is 0.014 lower than that of MBCNN*, which is a substancial improvement in similar visual tasks. Furthermore, as evident from Figure 6 and the visual results provided in Fig. 3 and Fig. 4 of the supplementary file, the results of MBCNN* still contain noticeable moiré residuals. In contrast, our method can remove these moiré patterns clearly. We will give more results in our website to demonstrate the visual improvements of our method over compared methods.
>
> > The two-branch structure has no rich interactions, but the network needs two stages for training.
>
> Thanks! In this work, we observe that the moiré patterns in different raw channels have different properties. Therefore, we extract color-mixed features and color-separated features, and propose channel modulation to fuse the two features together. Our solution is only a simple strategy to utilize the two kinds of features. A rich interaction may bring more gains, and we leave this as our future work. We find that two-stage training is better than one-stage training since our modulation operation requires a relatively comprehensive baseline feature representation. Fortunately, this does not influence the inference stage, which is more important in real applications. We will add these discussions in our final version.
>
> > The data set produced in this paper includes both raw domain and srgb domain, but only raw domain is used in the experiment, while most of the compared methods are designed based on the SRGB domain.
>
> We produced sRGB domain pairs due to two reasons.  First, most compared moiré removal methods are proposed with sRGB domain inputs. Our sRGB domain pairs can facilitate the comparison with these methods. Besides, we selected well-performing sRGB-domain demoiréing methods and retrained them with raw inputs. Although the performance is improved with raw inputs, they are still inferior to our methods. Second, our sRGB domain pairs can facilitate subsequent research related to sRGB-domain demoiréing and the raw-RGB moiré pairs can also facilitate inverseISP tasks. For our method, we only utilized raw inputs since our method is built upon our key observation that the moiré patterns in different raw channels are different.
>
> > Could the authors provide the results and comparison on the SRGB domain?
>
> Our channel modulation strategy was designed based on the observation that the moiré patterns in different raw channels are different.  However, moiré patterns in the sRGB domain do not exhibit such characteristics (refer to the comparison between the second and third rows in Figure R4). Our method is tailored to raw domain data and is not suitable with sRGB domain inputs. Therefore, we did not provide the results on the sRGB domain.

---

> ### Comment · Reviewer_piUN · 2023-08-18
> **Thanks for the rebuttal**
>
> The rebuttal has addressed my comments. I would like to keep my rating unchanged. I also suggest the authors include more latest video demoiréing methods for comparison.

---

> > ### Author Response · Authors · 2023-08-18
> >
> > Dear Reviewer,
> >
> > Thanks for your time and careful review!
> > We will incorporate more video demoiréing methods for comparison, including revising the image demoiréing methods to make them adapt to multi-frame based demoiréing.
> >
> > Best,
> >
> > Authors

---

### Official Review · Reviewer_irQe · 2023-07-01

**Soundness:** 3 good
**Presentation:** 4 excellent
**Contribution:** 3 good
**Rating:** 4
**Confidence:** 5

**Summary:**

This paper proposes an image and video demoiréing network tailored for raw inputs. It introduces a color-separated feature branch, and it is fused with the traditional feature-mixed branch via channel and spatial modulations. It builds the first well-aligned raw video demoiréing (RawVDemoiré) dataset and achieves state-of-the-art performance for both image and video demoriéing.

**Strengths:**

A raw demoireing dataset is proposed.
Experimental results demonstrate that the method achieves the best results on both raw video and image demoriéing datasets.

**Weaknesses:**

First of all, the network structure is not very novel. The modulation of channels is similar to channel attention mechanisms, and cross-channel and group modulation are also common practices. Secondly, the need for a large receptive field has been discussed in other articles on moiré pattern removal. Multiscale and large convolution kernels, as well as dilated convolutions, are also common approaches. From the experimental results in Table 3, it can be observed that the baseline (27.96) outperforms VDMoiré* (27.74) and vrt* (27.11). Therefore, I believe it is worth further exploring the effectiveness of the approach in details.

'Recaptured screen image demoiréing in raw domain' demoires in raw domain. I think it would be beneficial to transform it into a multi-frame version (e.g., by adding a PCD module) and then conduct a comparison. The paper can also compare some end-to-end methods that specifically handle raw domain images.

I am not clear about the significance of the experiments in Table 4. I believe that comparing the results of a single channel alone may not be as informative as the complete results. Therefore, I suggest moving this experiments from Table 4 to supplementary materials. Instead, it would be more helpful to include experiments that investigate the impact of loss functions and multi-scale constraints.


**Questions:**

see Weaknesses.

---

> ### Author Rebuttal · Authors · 2023-08-09
>
> Thanks for your nice summary and positive comments on our datasets and SOTA performance. In the following, we give detailed responses to your comments.
>
> > First of all, the network structure is not very novel. The modulation of channels is similar to channel attention mechanisms, and cross-channel and group modulation are also common practices. Secondly, the need for a large receptive field has been discussed in other articles on moiré pattern removal. Multiscale and large convolution kernels, as well as dilated convolutions, are also common approaches.  ...
>
> Our "channel modulation" operation is different from the traditional "channel attention" operation. In Table R2, we performed an ablation experiment by replacing our channel modulation with traditional channel attention. Despite channel attention utilizing more parameters (fully connected layers) to learn channel weights, its performance is inferior to that of our method. Fig. R5 presents the learned weights of channel modulation and channel attention. It can be observed that the channel attention weights are similar, which means there are no large differences. The main reason is that it needs a sigmoid layer to generate these coefficients. Different from it, our modulation coefficients are learned by setting them to be learnable parameters. The learned modulation coefficients have a large variance, which implies that different channels contribute very differently in the following.  Note that, our main novelty is jointly utilizing the color-separated and color-mixed features based on our observation that different channels have different moiré patterns in raw domain. **To our knowledge, there is no work giving this observation and no work exploring the complementariness of color-separated features and color-mixed features for image/video demoiréing. We believe this is a valuable contribution, as it poses new opportunities to solve the raw image (video) demoiréing problem.**
>
> As you mentioned, utilizing large receptive field for image demoiréing, such as multi-scale, large kernels, and dilated convolution, is a common strategy. In this work, we adopted a light-weight transformer-style convnet [6] to realize the spatial modulation task. This operation has only been utilized in visual recognition and our work is the first that transforms it to low-level vision tasks.  In this way, our spatial modulation not only increases the receptive field but also reduces the parameter and computation costs.
>
> As shown in Table 3 in the main paper, our CSMod strategy yields substantial gains on top of the baseline (e.g., nearly 0.8 dB increase in PSNR). We believe our work can inspire more works exploring the complementariness between color-separated and color-mixed features.
>
> Table R2: Comparison between our channel modulation and traditional channel attention.
> | Methods | PSNR | SSIM | LPIPS |
> | ---- | ---- | ---- | ---- |
> | channel attention | 28.492 | 0.9170 | 0.0987 |
> | channel modulation | **28.706** | **0.9201** | **0.0904** |
>
> > 'Recaptured screen image demoiréing in raw domain' demoirés in raw domain. I think it would be beneficial to transform it into a multi-frame version (e.g., by adding a PCD module) and then conduct a comparison.  The paper can also compare some end-to-end methods that specifically handle raw domain images.
>
> Thanks for your suggestion. We have incorporated the PCD module and fusion module into RDNet, transforming it into a multi-frame version. The results are presented in Table R3. Our method consistently outperforms RDNet on all the three metrics. Despite RDNet being a raw domain approach, it did not consider the moiré differences in raw domain channels and it merely adjusted the channel number of the first convolution layer to fit the raw inputs, without any other tailored operations for raw data. In contrast, our method is tailored to the distinctive channel distribution properties of raw domain moiré patterns, and our modulation strategy leverages the inherent advantages of raw domain data more effectively.
>
> Since there is no other raw domain demoiréing methods, we further utilized UNet, a raw domain method that performed well in the experiments of RDNet, and adapted it to a multi-frame version. The results of these experiments are presented in Table R3, and our method also outperforms the multi-frame UNet approach.
>
> Table R3: Comparison between multi-frame based raw video demoiréing.
> | Methods | PSNR | SSIM | LPIPS |
> | ---- | ---- | ---- | ---- |
> | Multi-frame RDNet | 26.970 | 0.9051 | 0.1176 |
> | Multi-frame UNet | 26.185 | 0.8917 | 0.1368 |
> | Ours | **28.706** | **0.9201** | **0.0904** |
>
> > About experiments that investigate the impact of loss functions and multi-scale constraints.
>
> Thanks for your suggestions. We will move Table 4 to the supplementary file and incorporate ablation experiments about loss functions and multi-scale constraints, as shown in Table R4. When the corresponding loss constraints were removed, there was a noticeable decrease in all evaluation metrics, which demonstrates the effectiveness of the hybrid loss functions and multi-scale constraints. Note that, the multi-scale constraints are widely utilized in benchmark demoiréing models, such as MBCNN [36], UHDM [26], and VDMoiré [3]. In addition, the perceptual loss is also included in all of our compared methods (except MBCNN, which utilizes a more advanced loss function). In summary, our method outperforms benchmark methods mainly due to our fusion of color-separated and color-mixed features and the proposed CSMod, which are our key contributions.
>
> Table R4: Ablation study about our loss functions and the multi-scale constraints.
> |Methods|PSNR|SSIM|LPIPS|
> |----|----|----|----|
> | w/o perceptual loss | 28.069 | 0.9166 | 0.1160 |
> | w/o color loss | 28.642 | 0.9169 | 0.0931 |
> | w/o multi-scale constraints | 27.455 | 0.9124 | 0.1041 |
> | Complete | 28.706 | 0.9201 | 0.0904 |

---

### Official Review · Reviewer_pajB · 2023-07-05

**Soundness:** 2 fair
**Presentation:** 3 good
**Contribution:** 2 fair
**Rating:** 4
**Confidence:** 4

**Summary:**

This paper addresses the problem of image/video demoireing through raw image data. The paper shows some observations, such as, the moire pattern in different color channels of the raw inputs have different properties, and pursues how to used this properties for raw image/video demoireing. It proposes to use channel and spatial modulations. To deal with this problem, the paper proposes a new dataset. The proposed algorithm achieves state-of-the-art performance.


**Strengths:**

The paper has some nice observations, such as the moire patterns in different color channels of the raw inputs have different properties. It tries to exploit this property and come up with a new algorithm using channel and spatial modulations. The algorithm achieves state-of-the-art performance.

**Weaknesses:**

Although the paper shows some nice observations, there is no in-depth analysis why these observations happens, e.g., why the R, G1, G2, and B channels have different moire patterns. Based on Figure 1, the moire shapes looks quite similar to me, only the intensities are different. My guess is that the grid for R, G1, G2, B are the same, only with a pixel shift in locations, and the color filters will filter out the light for different colors, and thus have different intensities, but I am not exactly sure. The paper should provide more insight into this aspect.

The paper put a lot of efforts into building a new video raw moire dataset, and it is actually the best selling point of this paper from my point of view. Why is this necessary?  If one want to deal with raw image/video demoireing, is not it easier to directly convert the sRGB images to RAW images, and spend moire efforts on designing better architectures?

I am not quite convinced by the novelty in the network design, it is simply follow the pipeline of feature extraction, alignment, temporal fusion and reconstruction. It combines with a few modules, such as PCD alignment, but I am not convinced if this is novel.

**Questions:**

none

**Limitations:**

see weaknesses

---

> ### Author Rebuttal · Authors · 2023-08-09
>
> Thanks for your nice summary and positive comments on our novelty and contributions. In the following, we give detailed responses to your comments.
>
> > Although the paper shows some nice observations, there is no in-depth analysis why these observations happens, e.g., why the R, G1, G2, and B channels have different moiré patterns. Based on Figure 1, the moiré shapes looks quite similar to me, only the intensities are different. My guess is that the grid for R, G1, G2, B are the same, only with a pixel shift in locations, and the color filters will filter out the light for different colors, and thus have different intensities, but I am not exactly sure. The paper should provide more insight into this aspect.
>
> Thanks for your careful review. We would like to point out that the moiré shapes in R, G1, G2, and B channels are different. In Fig. R4, we present the zoom-in version of the moiré patterns in different raw channels for better observation. For the exampled image, its R channel only contains vertical and horizontal stripes with the same regular scale. In contrast, its G2 channel contains fine-scale moiré patterns besides the regular scale ones. To make this clearer, we further present the DCT spectrum of the four channels. The moiré patterns usually appear as nearly periodic stripes, which implies that the moiré patterns will be represented as strong peak spots in the DCT spectrum. As shown in the second row of Fig. R4, the R and G1 channels only have peak spots at the left-up corner, which represent coarse-scale moiré patterns and this is consistent with the observation in the image domain. Meanwhile, the G2 and B channels have peak spots at left-up, right-up, and left-down corners. It means that the two channels have both coarse-scale and fine-scale moiré patterns, which is consistent with the observation in the image domain. Note that, all the images are normalized (the DCT spectrum is also calculated from normalized image) to avoid the effects of different intensities of different CFA channels.
>
> The next important question is why they have different moiré patterns. We would like to point out that the moiré patterns are affected by the screen, camera, viewpoint, and image content. As you mentioned, there is a pixel shift in locations across the Bayer pattern channels in the raw domain. In other words, the viewpoints of the R, G1, G2, and B channels are different. During data acquisition, even one-pixel shift can have a significant impact on the period and shape of moiré patterns, rather than being just a one-pixel displacement. On the other hand, with the same camera, screen, and viewpoints but different contents, the generated moiré patterns are also different. For the R, G1, G2, and B channels of the same image, their contents are also different to some extent (due to different spectrum responses of the color filter). Therefore, the moiré patterns of the four channels are different. In our final version, we will add these deep analysis for our observations.
>
> > ...Why not directly convert the sRGB images to RAW images, and spend moiré efforts on designing better architectures?
>
> We would like to point out that it is difficult to inverse a clean RGB image into raw domain accurately since the ISP process is nonlinear and the raw image has wider bit depth than that of the sRGB image. If the RGB image contains moiré patterns, it will be more difficult to reproduce the moiré patterns captured in the original raw domain.  In Fig. R4, we present the DCT spectrum of the moiré images across different channels in both raw and sRGB domain. As mentioned in the response to your first question, the peak spots in the DCT spectrum represent the moiré patterns. The four channels in the raw domain have different peak spots, which means the moiré distributions are also different. Meanwhile, the DCT spectrum of the R, G, and B channels of the RGB image are similar. The reason is that the complex ISP process mixes these moiré patterns. Then we perform inverseISP (using the implementation provided by CycleISP) on the moiré RGB image, and the DCT spectrum of the inversed raw image are presented on the fourth row. It can be observed that they still have similar peak spots. In other words, the distinct moiré patterns in different raw channels cannot be recovered. Therefore, utilizing the real captured raw images for demoiréing network training is better than utilizing inversed (synthesized) raw images. In addition, even for denoising, utilizing real captured raw is also better than using synthesized raw, as shown in [29].  Utilizing inverseISP for dataset generation is a good topic for further exploration but our work did not focus on this point.  Our constructed dataset not only benefits the network training but also provides a real test set to evaluate different demoiréing methods.
>
> >I am not quite convinced by the novelty in the network design, it is simply follow the pipeline of feature extraction, alignment, temporal fusion and reconstruction. It combines with a few modules, such as PCD alignment, but I am not convinced if this is novel.
>
> The pipeline of feature extraction, alignment, temporal fusion and reconstruction is our baseline other than our contribution. The main novelty of this work is the design of channel modulation and spatial modulation, which are devised based on our observations that moiré images in the raw domain channels have different properties. If we utilize traditional convolution to recover shared structures, the moiré patterns in different channels are also spread. Therefore, we propose to use the complementariness between color-separated features and color-mixed features. To our knowledge, in the literature, there is no work giving this observation and no work exploring the complementariness of color-separated features and color-mixed features.  We believe this is a valuable contribution, as it poses new opportunities to solve the raw image (video) demoiréing problem.

---

### Official Review · Reviewer_AQcs · 2023-07-06

**Soundness:** 3 good
**Presentation:** 3 good
**Contribution:** 3 good
**Rating:** 6
**Confidence:** 3

**Summary:**

This paper proposed an image and video demoiréing network tailored for raw inputs.  The authors constructed the first real raw video demoiréing dataset with temporal and spatial aligned pairs and designed a network tailored with two branches including one branch to share feature enhancement and the other branch to modulate different color groups. Experimental results demonstrate that the proposed method achieves the best results on both raw video and image demoriéing datasets.

**Strengths:**

+This paper first explores raw video demoiréing and constructs the first RawVDemoiré dataset.
+In this paper, several strategies are designed to solve the problem of time synchronization and spatial alignment between the original image and the shot image in data acquisition.
+The experimental results presented in this paper demonstrate the effectiveness of the collected dataset to improve the performance of the demoiréing network.

**Weaknesses:**

-Although the experimental results demonstrate the effectiveness of the collected dataset, compared with UHDM*, the performance of the proposed network has no significant advantages. Neither numerical comparison nor qualitative results can clearly show the advantages of the network.
-This paper does not analyze the performance limits of the proposed network, that is, the performance and advantages in the face of extremely severe moire patterns.

**Questions:**

-Are the image observations of different raw image channels in Figure 1 representative?

**Limitations:**

Yes, the authors have discussed the limitations and potential negative societal impact of their work.

---

> ### Author Rebuttal · Authors · 2023-08-09
>
> Thanks for your nice summary and positive comments on our datasets. In the following, we give detailed responses to your comments.
> > -Although the experimental results demonstrate the effectiveness of the collected dataset, compared with UHDM*, the performance of the proposed network has no significant advantages. Neither numerical comparison nor qualitative results can clearly show the advantages of the network.
>
> We would like to point out that our method does outperform UHDM* in both qualitative and quantitative measurements. Due to the page limits, our main paper only provides two visual comparisons. In Fig. 6, although the moiré removal performance of our method is similar to that of UHDM*, our method works better in color reproduction. In Fig. 7, our method can remove the moiré patterns clearly but there are still obvious moiré patterns in the result of UHDM*. In addition, the visual comparison results in Fig. 3 and Fig. 4 of the supplementary file also demonstrate that the results of UHDM* have color cast and the reconstructed edges are kind of blurry. In contrast, our method can remove the moiré patterns clearly and recover sharp edges and correct colors.  For comprehensive comparison, we further present two examples in Fig. R1.
>
> For quantitative comparison (Table 1 and Table 2 in the main file, which is also summarized in the following Table R1), our method outperforms UHDM* for both image and video demoiréing. Since it is difficult to use one measurement to evaluate the demoiréing and detail reconstruction performance, we use PSNR, SSIM, and LPIPS for comparison. Among these metrics, LPIPS is more effective at reflecting the quality of moiré pattern removal. For image demoiréing, our method outperforms UHDM* by 0.0064 in terms of LPIPS, which is considered a substantial improvement in relevant visual tasks. For video demoiréing, our method outperforms UHDM* with a large margin. One reason is that we utilized multiple frame correlations. The other important reason is that the moiré patterns and color distortion in the video demoiréing dataset are more heavier than that in image demoiréing dataset, and our method is much better in dealing with these distortions than UHDM*.
>
> Table R1: Comparison between our method and UHDM*.
> | Category | Methods | PSNR | SSIM | LPIPS |
> |  ----  | ----  |  ----  | ----  |  ----  |
> | Image Demoiréing | UHDM* | 27.078 | 0.9302 | 0.0812 |
> | Image Demoiréing | Ours | **27.261** | **0.9346** | **0.0748** |
> | Video Demoiréing | UHDM* | 25.642 | 0.8792 | 0.1232 |
> | Video Demoiréing | Ours | **28.706** | **0.9201** | **0.0904** |
>
> > -This paper does not analyze the performance limits of the proposed network, that is, the performance and advantages in the face of extremely severe moiré patterns.
>
> Thanks for your suggestion. When the image has high color saturation and the moiré pattern contrast is strong, our method may cannot remove the patterns clearly. Fig. 1 in the main paper belongs to this case. Our method can remove the moiré patterns to some extent, while some compared methods (such as RDNet and MBCNN) failed in dealing with this hard example. Fig. R2 also gives an example. When dealing with this kind of challenging example, our method still works the best. In the final version, we will introduce the discussion and give more examples about this limitation.
>
> > -Are the image observations of different raw image channels in Figure 1 representative?
>
> Yes, our observation that different channels have different moiré patterns is representative. In Fig. R3, we provide eight examples of the moiré frames and give their DCT spectrum. The moiré patterns usually appear as nearly periodic stripes, which implies that the moiré patterns will be represented as strong peak spots in the DCT spectrum. It can be observed that the DCT spectrum of different raw channels in the same image are different.  This is consistent with our observations in the image domain that different raw channels have different moiré patterns.

---

> ### Comment · Reviewer_AQcs · 2023-08-19
>
> Thanks to the authors for their response, the rebuttal has addressed my concerns and I will not change the original rating.

---

> > ### Author Response · Authors · 2023-08-21
> >
> > Dear Reviewer,
> >
> > Thanks for your careful review and support to our work!
> >
> > Best,
> >
> > Authors

---

### Author Rebuttal · Authors · 2023-08-10

Dear Reviewers,

We sincerely thank you all for your careful review and helpful suggestions, which significantly helped to improve our paper. In the following, we give separate responses to your comments one by one. The figures used for further illustration are presented in the pdf file, and they are named as Figures. R1-R6.

Best Regards,

Authors of 6552

---

### Decision · Program_Chairs · 2023-09-21

**Decision:**

Accept (poster)

**Comment:**

This paper presents a new approach for video demoireing. It observes that different channels have different moire patterns, and proposes an approach that leverages this fact. Reviewers also appreciated that it introduces a new video dataset for the task. While there was some concern about limited novelty, in general the reviewers agreed that the solution was validated by experimental results, and that this work may inspire future research in this field.